



# Introducing *GloRiSe* – A global database on river sediment composition

Gerrit Müller[1], Jack J. Middelburg[1], Appy Sluijs[1]

[1]Department of Earth Sciences, Utrecht University, Utrecht, The Netherlands

*Correspondence to*: Gerrit Müller (g.muller@uu.nl)

**Abstract.** Rivers transport dissolved and solid loads from terrestrial realms to the oceans and between inland reservoirs, representing major mass fluxes on Earth's surface. The composition of river water and sediment provides clues to a plethora of earth and environmental processes, including weathering, erosion, nutrient and carbon cycling, environmental pollution, reservoir exchange and tectonic cycles. While there are documented, publicly available databases for riverine dissolved and

suspended nutrients, there is no openly accessible, georeferenced database for riverine suspended sediment composition. Here, we present a globally representative set of 2828 suspended and bed sediment compositional measurements from 1683 locations around the globe. This database, named Global River Sediments (*GloRiSe*) version 1.0, includes major, minor and trace elements, along with mineralogical data, and provides time-series for some sites. Each observation is complemented by metadata describing geographic location, sampling date and time, sample treatment and measurement details, which allows

for grouping and selection of observations, as well as for interoperability with external data sources and improves interpretability. Information on references, unit conversion and references make the database comprehensible. Notably, the close to globe-spanning extent of this compilation allows to derive data-driven, spatially resolved global-scale conclusions about the role of rivers and processes related to them within the Earth-system.

*GloRiSe* version 1.0 can be downloaded from Zenodo  (doi:10.5281/zenodo.4447435, Müller et al., 2021) and

https://github.com/GerritMuller/GloRiSe.git (where updates with adapted version numbers will become available), along with a technical documentation and an example calculation in the form of a MATLAB script, that calculates the sediment-flux weighted major element composition of the annual riverine suspended sediment export to the ocean and related uncertainties.

## 1 Introduction

Rivers are major drivers of material transport, processing and deposition on Earth's surface (Martin and Meybeck, 1979; Milliman and Farnsworth, 2011; Viers et al., 2009). This makes them important for ecosystem functioning within the respective catchment and in the coastal oceans or lakes they feed (Allan and Castillo, 2007; Meybeck, 1982). Flora and fauna (including humans) often live along rivers and at their mouths, because they provide freshwater, nourishment, energy and effective



possibilities of transportation (Allan and Castillo, 2007). Increasing human populations have led to environmental pollution
and disturbance of natural processes (Martin and Meybeck, 1979; Nienhuis et al., 2020; Vörösmarty et al., 2010). For instance,
increased nutrient input related to land-use change and agricultural activities (Beusen et al., 2016; McDowell et al., 2020b;
Weigelhofer et al., 2018), acidification and heavy metal contamination (Kumar et al., 2019), harm biodiversity and people
around the globe, and damming increasingly affects riverine fluxes, processes and ecosystems (Mulligan et al., 2020;
Vörösmarty et al., 2010).

As transporters of solutes and solids, rivers play a major role in terrestrial weathering and erosion and thereby in the global
carbon cycle, mediating atmospheric greenhouse gas concentrations and thus climatic stability of the Earth-system on geologic
time-scales (Berner, 2003; Caves Rugenstein et al., 2019; Isson et al., 2020). Riverine suspended sediment fluxes of organic
carbon (Berner, 1982; Hilton and West, 2020), phosphorous (Berner, 1999; Froelich et al., 1982) and biogenic silica (Conley,
2002) are considered dominant terms in their global budgets and a similar importance was proposed for the riverine particulate
fluxes of calcium (Gislason et al., 2006), strontium (Jones et al., 2012) and inorganic carbon (Middelburg et al., 2020).
Moreover, significant amounts of divalent cations weakly bound to the negatively charged (clay) mineral surfaces are
transported downstream with the suspended particles and this complicates estimates of weathering rates (Cerling et al., 1989;
Tipper et al., 2021).

Superimposed on the lithological composition of the catchment, the interplay of terrestrial weathering and erosion define the
composition of water and sediment input to the rivers and the magnitude of the corresponding fluxes. Therefore, dependencies
of weathering and erosion can be studied through the rivers composition in comparison to environmental variables (Gaillardet
et al., 1999; Hartmann et al., 2014b; Romero-Mujalli et al., 2019). Extensive datasets describing riverine hydro-chemistry
(GLORICH, Hartmann et al., 2014a; Nitrogen & phosphorous, McDowell et al., 2020a) and environmental variables  (e.g.
HydroBasins, Linke et al., 2019) have recently been established. Databases on water and sediment discharge, have been
developed (GEMS-GLORI, Meybeck and Ragu, 1997; Milliman and Farnsworth, 2011; *Land2Sea* Peucker-Ehrenbrink, 2009).
In contrast, established major and trace element budgets of river suspended sediment (Martin and Meybeck, 1979; Savenko,
2007; Viers et al., 2009) are not easily (if at all) accessible, comprehensible or fully traceable, leaving much of their potential
unexplored. To our knowledge, no current database summarizes the mineralogical and petrographic composition of global
river sediments.

To fill this gap we compiled published (108 articles) – mostly peer-reviewed (104) – major, minor and trace element,
mineralogical and petrographic data of 2828 suspended and bed sediment samples taken at 1683 different world-wide locations
between 1874 and 2016 Complementary metadata provides a spatio-temporal context and facilitates traceability of each
datapoint, grouping/selection operations within and interoperability of this database, named *Global River Sediment* (*GloRiSe*).
Practically, this data may be used for e.g. (spatial) statistical modelling and model testing, to complement local or regional
datasets, to explore and compare time-series at different locations, to screen potential for field studies, to assess anthropogenic
pollution or to characterize the material continuously transported to the global ocean and fresh-water reservoirs. Notably, the
globe-spanning extent of the database allows to derive data-driven global-scale conclusions about the role of rivers and



processes related to them within the Earth-system. In this article, we explain the derivation, harmonization and structure of the database, comment on its extent and limitation, and complement this by an example application, the calculation of the major

element composition of the annual riverine suspended sediment export to the ocean.

**2 Data collection**

Data was collected during a literature survey between March and September 2020 of studies with local to global extent. The survey aimed at suspended sediment samples, being best representative for two potential fields of application: (a) the integrated chemical weathering history of the catchment (von Eynatten, 2004; Guo et al., 2018; He et al., 2020; Nesbitt and Young, 1982)

and (b) the characterization of terrestrial riverine material transport and export (Ludwig et al., 1996; Milliman and Farnsworth, 2011). Riverbed sediments were primarily considered where no information on suspended sediment samples was found (especially in Africa) and marked specifically. If available, grain size information was included as well (sieve and/or filter pore size, average grain size, sand/silt/clay percentage). The fine fraction, i.e. clay and silt sized particles, of present and past riverine deposits is often considered a compositional tracer for riverine suspended load (Fedo et al., 1995; Garzanti et al., 2015; Guo

et al., 2018; Nesbitt and Young, 1982). However, there is an effect of hydrodynamic sorting during deposition, affecting sediment composition (Bouchez et al., 2011; von Eynatten et al., 2012; Galy et al., 2007; Garzanti et al., 2010, 2011; Horowitz and Elrick, 1987; Lupker et al., 2011). Dissolution, precipitation, particle break-down and resuspension along the river further alter their chemical composition (Cole et al., 2007; Ensign and Doyle, 2006; Lupker et al., 2012; Nakato, 1990; Négrel and Grosbois, 1999; Papanicolaou et al., 2008). These findings need to be considered in the selection of *GloRiSe* data for a specific

application. Because included samples should at least resemble the complete inorganic composition of suspended matter, no studies were included, in which samples were decarbonated before analysis (e.g., Bayon et al., 2015; Liu et al., 2007, 2012). Other sample treatments, such as oxidation, fusion or digestion were noted.

Previous compilations served as a starting point of our survey (Viers et al., 2009), or were incorporated (Martin and Meybeck, 1979 & Meybeck and Ragu, 1997), depending on their content. Studies were selected, that report either inorganic major and

minor element composition (Si, Al, Ca, Mg, K, Na, Fe, Mn, Ti, P, S, C, Loss on ignition – together 2412 observations) and/or a (semi-)quantitative mineralogical phase analysis (876 observations). Organic C, P, N and trace-elements were only added when they were reported along with the major element or mineralogical data, implying that there is ample room to further expand *GloRiSe* using published data. Nevertheless, 1906 observations of selected trace-elements and  700 observations of organic C were collected. Organic P and N are highly under-represented. When instantaneous water discharge and/or

suspended sediment concentration and/or solution properties (T, pH, alkalinity, $Si(OH)_4$, DIC, DOC, $HCO_3^-$, $SO_4^{2-}$, $Cl^-$, $Ca^{2+}$, $Mg^{2+}$ $Na^+$, $K^+$, Calcite-saturation) were reported in the same study, these were also added. Doublings with entries of other databases were not checked. Units were properly conversed (Appendix A). Studies were furthermore only included, if geographic coordinates were given or could be assigned using © Google Earth 2020 and given maps and/or site-names. For spatial averages (21 samples), coordinates in the center of the location distribution were chosen (range < 1 to ~ 5 °





latitude/longitude). Country and region of the measurement were also noted following details given in the specific study. The closest information on sampling date is given for each observation, which can range in resolution from some years to minutes, but is the day or month in most cases. Seasonal (26) and annual (143) averages were also included, if the original measurements were not accessible, which is especially the case for older publications (before ~ 2000).

## 3 Database structure

The structure of the database employs that of the complementary GLORICH database (Hartmann et al., 2014a) listing variables in columns and identifiers, metadata and observations in rows. It consists of 6 separate tables, in which samples are linked via a unique Sample_ID, which in turn, can be related to a Location_ID that is similar for all observations from exactly the same site. Locations that are situated within the same main basin were assigned the same Basin_ID, that allows to group observations in terms of catchment without further processing using geographic information systems (GIS). A main basin delineates

everything that drains into the same river that is tributary only to the ocean (or a lake for endorheic drainage). The Rep_ID was introduced, allowing to distinguish samples that are representative for the export to the ocean in terms of sampling position in the lower course of the main stem river but before significant marine influence, from upstream and tributary observations or endorheic drainages. Marine influence was assessed using tidal maps (Matthews, 2014), coastal landforms (© Google Earth 2020) and, where available, information on salinity gradients or assignment of freshwater endmembers. The Rep_ID also

allows to distinguish observations during storm- or flood events, although the assignment of this relied on information provided in the source article.

All information necessary to identify and retrace a sample, such as the IDs assigned by us, its original ID used in the original study, sampling date and references are stored within a separate table. Information on each variable is given in each file and a technical documentation and explanatory script can be downloaded together with the database.

## 4 Extent and limitations


The database covers a representative set of downstream observations from all over the globe except for Antarctica, but leaves larger gaps in upstream and endorheic areas (Figure 1). Further gaps are identified in parts of North and Central Canada, Western USA, Central America, Western South America, Brazil, Northwestern Africa and Oceania (Figure 1). Although a few observations from South West Greenland and Central East Greenland are included, they are unlikely representative for

Greenland sediment discharge as a whole. Upstream measurements are also not available for many parts of the world.

In contrast to the globally representative spatial coverage, temporal coverage is very low, i.e. there are few time series included. This is mostly because there are very few time series and concentration-discharge relationships available in literature.

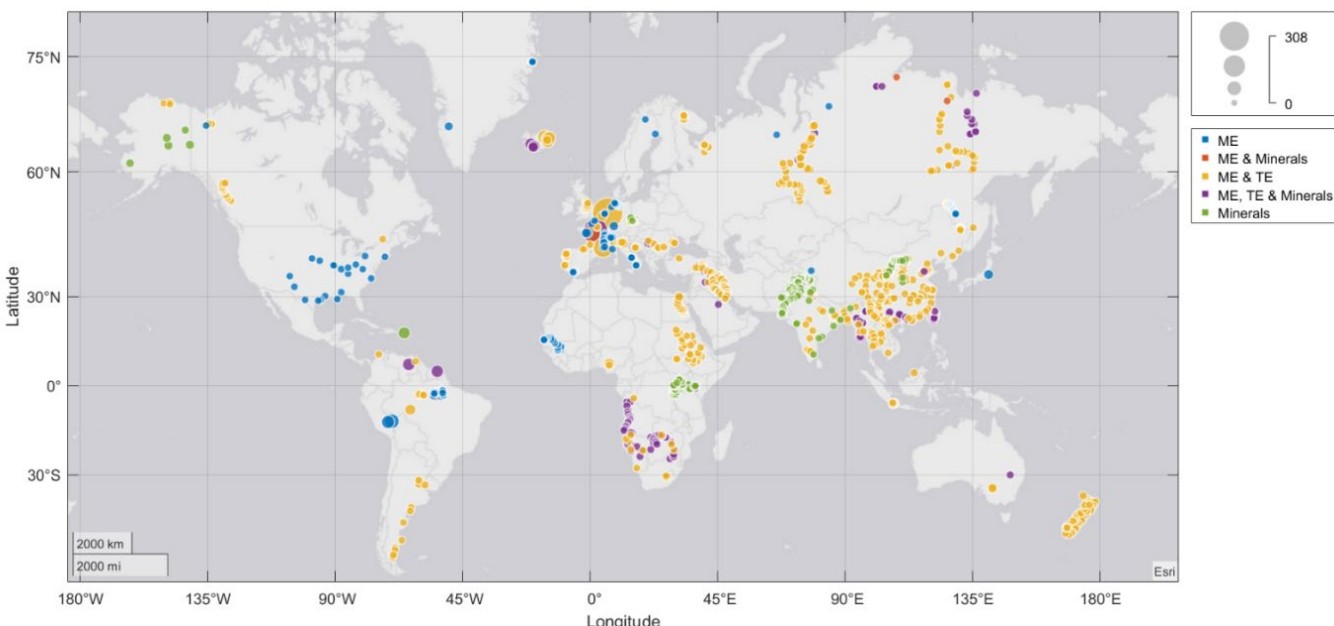

**Figure 1 Spatial extent of the database. Dot size indicate the number of observations available per sample. Dot colour indicates what kind of data is available at that location (ME: major & minor elements, TE: trace elements).**

The number of available variables differs between the samples in respect to the original purpose of data acquisition. Consequently, depending on the users aim, the number of suitable observations may decrease drastically, i.e., the more variables needed, the less datapoints are available. This problem aggravates if the database is to be combined to other (point) data sources, because the range of intersecting locations might be small or require a decrease in spatial resolution by e.g. rounding coordinates or integrating over geographic shape layers. The database does obviously not (yet) cover the whole overwhelming amount data on river sediment composition in the literature but is intended to be further expanded. We therefore not only ask users to provide feedback but also to contribute to extension of the database by sending us (also unpublished) suitable data with proper metadata to be included.

## 5 Possibilities and perspectives

To demonstrate some important possibilities and perspectives of the database, we provide an example MATLAB script for the calculation of the average major element composition of the global riverine suspended sediment export, including the combination with external data sources (Milliman and Farnsworth, 2011) and detailed comments. In this example, we exclude bedload samples coarser than very fine sand (grain sizes > 125 μm or unknown) and split up samples according to their observation type into single measurements and spatial averages (I), seasonal averages (II) and annual averages (III). Rep_ID





and Basin_ID are used to select samples from river main stems that represent the basins sediment discharge to the ocean. For each basin in the subsets (I) and (II), mean and median major element concentrations are calculated, representing annual averages for each set. All subsets are then recombined and an annual mean and median are calculated for each basin. This data

is then merged with annual average sediment discharge values based on the river names (Milliman and Farnsworth, 2011) to calculate basin-wise fluxes of each major element oxide. We then calculated the global mean and median concentrations of each oxide from the basin-wise mean and median, respectively, and the sediment-flux-weighted (sfw) mean. These estimates are based on samples, that can be considered spatially representative and probe ~ 35 % of the total global sediment flux considering 19.1 Pg/a (Beusen et al., 2005; Milliman and Farnsworth, 2011).

Temporal variability of element concentrations in individual rivers is accompanied by an even larger temporal variability of sediment fluxes (Clark et al., 2017; Cohen et al., 2014; Eberl, 2004; Eiriksdottir et al., 2008; van der Perk and Vilches, 2020; Rousseau et al., 2019) and, thus, imposes a major uncertainty on the basin-wise averages, for which often only very few measurements are available (down to 1). To quantify this uncertainty, five to seven sites (depending on the element) within the *GloRiSe* database, were selected for which time-series spanning at least 10 months are available (along Amazon, Orinoco,

Rhine, Loire, Rhône and Kuji). We then calculated the sfw mean major element concentrations and determined the maximum difference and the sample standard deviation relative to this sfw mean concentration ($SD_{sfw,t}$) from each time-series. The mean for extreme values is 0.3 – 29.3 wt%, while the mean $SD_{sfw,t}$ of all series range from 0.1 to 8.6 wt% (Table 1). We propagated the mean $SD_{sfw,t}$ through the standard errors of the global averages under the worst case assumption, that only one measurement is available instead of a time-series for each location. The error estimate is therefore regarded as an upper limit and because

many of the rivers available as time-series are comparably small and carbonate-rich (Rhine, Loire, Rhône), potentially implying shorter response time to excursions of the flow regime. Furthermore, we inherently assume in our error estimate, that event-scale variability is within the month-scale variability, which should be subject to future research. We ignore measurement errors, suggesting them to be much smaller than seasonal variation. Uncertainties of sediment fluxes in the weighing procedure are also neglected, because we suggest them to be of minor importance compared to inter-basin

differences, spanning several orders of magnitude (Cohen et al., 2014; Milliman and Farnsworth, 2011).

The median differs from -1.4 to 4.9 wt% from the mean and should be preferred for all oxides except for $SiO_2$ and MnO because of their log-normal distribution. We applied this to our global sfw mean, which may be more representative for the global sediment export. Results for major and minor elements sum to 75 – 85 %, the rest is accounted for by organic matter, degassing during sample preparation (e.g. $CO_2$ and $H_2O$ degassing upon fusion) and uncertainty. To estimate the fraction of

major cations sorbed to negatively charged mineral surfaces instead of being truly incorporated into the solid phases, we reproduced and utilized a published linear relationship between clay-mineral controlled molar Al/Si ratios and cation exchange capacity (CEC), along with estimates of the average major element composition of this sorbed pool (Tipper et al., 2021). With a global average molar Al/Si ratios of 0.373, we arrive at an average CEC of 31.78 meq/100 g, implying 0.72 wt% Ca, 0.11 wt% Mg, 0.02 wt% Na and 0.03 wt% K of our sfw mean estimates to be derived from the sorbed pool.



Table 1 holds the results normalized to a total of 100 % along with uncertainties, the quantification of the sorbed pool and previous estimates for global average major element composition, while unnormalized and intermediate results are available through the tutorial script and can directly be downloaded with the script.

Our global mean estimates of $SiO_2$, MgO and $Na_2O$ appear to be higher than previous estimates from literature, while $Al_2O_3$ and $Fe_2O_3$ mean values are lower (Table 1). CaO, MnO, $K_2O$, $TiO_2$ and $P_2O_5$ are within the range of the selected literature

estimates. Part of the similarity to the results of Viers et al., (2009) and Martin and Meybeck (1979) originates from common data sources: We used the compilation of Viers et al., (2009) as a starting point for our survey and the data of Martin and Meybeck (1979) is fully included into *GloRiSe*. However, Viers et al., (2009) used the mean of main stem and tributary measurements, while Martin and Meybeck (1979) report the median of downstream mainstem locations. The derivation of the values presented in Savenko (2007) is not retraceable so we cannot evaluate possible common data sources. High sample

standard deviations exceeding the actual concentration for some alkali and alkaline earth elements (Table 1) again stress the heterogeneous and time-variant nature of riverine suspended sediment composition (Clark et al., 2017; Eberl, 2004; Eiriksdottir et al., 2008; van der Perk and Vilches, 2020; Rousseau et al., 2019).

Compared to the continental crust (Rudnick and Gao, 2013), $Na_2O$, $K_2O$ and CaO are depleted in riverine suspended sediments, while $Al_2O_3$, $Fe_2O_3$ and $TiO_2$ are enriched. This can be explained by terrestrial low-temperature weathering, which transforms

pristine mineral phases (e.g. feldspar, mafic minerals and calcite) into dissolved load and secondary phases (Putnis et al., 2014; Ruiz-Agudo et al., 2016). Well-soluble elements (Na, K, Mg and Ca) will preferably be transported as dissolved load, leaving rather insoluble elements (Al, Fe, Ti) enriched in the weathering product (Gaillardet et al., 1999; Garzanti et al., 2014a; Middelburg et al., 1988; Nesbitt and Young, 1982; Stroncik and Schmincke, 2001). Consequently, the unweathered source rock is relatively rich in mobile elements and also more reactive (Brantley et al., 2008; Lasaga, 1984). To quantify the relative

contribution of weathered and unweathered material to the composition of fine-grained sediment samples, reflecting weathering intensity within the sediments source area, different chemical weathering indices have been developed (Fedo et al., 1995; Gaillardet et al., 1999; Garzanti et al., 2013, 2014b; Harnois, 1988; Nesbitt and Young, 1982; Parker, 1970). Most of them are based on the relative concentrations of mobile to quasi-insoluble elements (Fedo et al., 1995; Garzanti et al., 2014b; Harnois, 1988; Nesbitt and Young, 1982; Parker, 1970), each involving different elements and related pitfalls. Other indices

rely on the distribution of elements between solution and particles (Gaillardet et al., 1999; Garzanti et al., 2013). For the current application, where we lack sufficient information on dissolved concentrations and do not expect a strong diagenetic imprint, but need to account for carbonate- and phosphate-related CaO, we use the chemical index of alteration CIX (Garzanti et al., 2014a):

$$CIX = 100 * Al_2O_3/(Al_2O_3 + Na_2O + K_2O) \qquad (1)$$

We neglect the contributions of sorbed $Na_2O$ and $K_2O$, because their magnitude is small compared to solid concentrations (Table 1). High CIX values imply a large contribution of weathered material, while a low CIX, similar to that of parent rocks, implies a substantial contribution of unweathered material. As riverine suspended sediment composition represents a mixture



of eroded pristine material (parent rocks) and weathering products with river-internal processing, our lower global average CIX values imply the exported material to be less weathered and hence more reactive than anticipated. Note, that the higher

values from literature are not sediment-flux weighted. This observation is statistically significant with respect to the propagated error of the CIX (± 0.1 wt%) and is explainable by the rivers included in each dataset: Cold-climate rivers exhibit lower chemical weathering rates in their catchment (Hartmann et al., 2014b) and mountainous rivers are characterized by steeper terrains, higher erosion rates and less soil formation, hence chemical weathering (Milliman and Syvitski, 1992; Patton et al., 2018). The CIX calculated from the compilation of Savenko (2007) is rather in the range of our estimates, which may be

explained by the inclusion of more arctic Russian rivers compared to the other literature estimates, as deduced from the literature cited along. However, the CIX does not include $SiO_2$, which is up to 9 wt% lower in Savenko (2007), than in our study. These explanations are consistent with the marked decrease in our sfw mean CIX compared to the mean and median estimates, because the former is largely influenced by a few large rivers draining areas of high chemical weathering intensities, i.e. Amazon, Ganga-Brahmaputra, Changjiang, Congo, Irrawady, Orinoco, Magdalena, Mekong and Godavari together already

deliver ~ 20 % of the global sediment flux and South East Asian drainages contribute as much as 60 % of the global sediment budget (Milliman and Farnsworth, 2011).

**Table 1** Global average major element composition of riverine suspended matter discharged to the ocean in weight % relative to total suspended matter and normalized to 100 % excluding organic matter and oxides often lost during preparation (e.g. $CO_2$ and $H_2O$) The mean, median and sediment flux weighted average of this study are compared to previous estimates from literature. T

denotes the total concentration (ferric + ferrous Fe or organic + inorganic P). CIX is a chemical alteration index calculated on a molar basis (Garzanti et al., 2014; Eq. (1)). The error estimate belongs to the flux-weighted mean and does account for sampling-time induced uncertainty and sample standard deviation and number of samples (N). Sorbed load was calculated based on published CEC, Al/Si ratios and mean fraction of Ca, Mg, Na and K in the sorbed pool (Tipper et al., 2021).

| | $SiO_2$ | $Al_2O_3$ | $Fe_2O_3T$ | MnO | CaO | MgO | $K_2O$ | $Na_2O$ | $TiO_2$ | $P_2O_5T$ | CIX |
|---|---|---|---|---|---|---|---|---|---|---|---|
| **Mean this study** | 64.5 | 15.9 | 7.3 | 0.2 | 3.5 | 3.3 | 2.2 | 1.9 | 0.9 | 0.3 | 73.74 |
| **Median this study** | 68.2 | 15.9 | 6.7 | 0.1 | 2.3 | 1.9 | 2.3 | 1.6 | 0.8 | 0.2 | 74.42 |
| **Sediment flux-weighted mean this study (suggested)** | 62.8 | 19.9 | 6.7 | 0.1 | 3.1 | 2.4 | 2.8 | 1.1 | 0.8 | 0.1 | 77.45 |
| **Sorbed load in sfw mean** | | | | | 0.72 | 0.11 | 0.03 | 0.02 | | | |
| **Sample SD** | 15.0 | 4.6 | 3.4 | 0.1 | 3.7 | 10.7 | 0.9 | 1.2 | 0.5 | 0.3 | |
| **Maximum basin-wise uncertainty from time-series** | 29.3 | 11.0 | 4.3 | 0.3 | 10.5 | 1.2 | 1.4 | 0.6 | 0.4 | 0.6 | |
| **Mean basin-wise uncertainty from time-series** | 8.6 | 5.2 | 1.5 | 0.1 | 2.9 | 0.5 | 0.6 | 0.2 | 0.2 | 0.1 | |
| **N (basins)** | 156 | 174 | 174 | 173 | 174 | 173 | 171 | 169 | 144 | 145 | |
| **Uncertainty of global average** | 3.22 | 0.65 | 0.45 | 0.02 | 0.80 | 1.45 | 0.11 | 0.28 | 0.07 | 0.08 | 0.1 |
| **Viers et al. 2009 (mean)** | 58.0 | 19.9 | 10.0 | 0.3 | 4.4 | 2.5 | 2.5 | 1.2 | 0.9 | 0.4 | 78.98 |
| **Martin & Meybeck 1979 (median)** | 61.1 | 20.2 | 7.8 | 0.2 | 3.4 | 2.2 | 2.7 | 1.1 | 1.1 | 0.2 | 78.03 |
| **Savenko 2007 (unknown)** | 58.6 | 19.8 | 8.7 | 0.2 | 4.4 | 2.9 | 3.1 | 1.3 | 0.8 | 0.2 | 74.93 |
| **Continental Crust (Rudnick & Gao 2003)** | 66.6 | 15.1 | 4.1 | 0.1 | 4.2 | 2.3 | 3.2 | 3.6 | 0.5 | 0.2 | 62.32 |



## 6 Database and code availability

*GloRiSe* 1.0 can be downloaded in form of Excel-sheets (.xlsx), Comma-separated vectors (.csv) or MATLAB Data format (.mat) from Zenodo (doi:10.5281/zenodo.4447435, Müller et al., 2021) and https://github.com/GerritMuller/GloRiSe.git along with the technical documentation. MATLAB scripts of the presented calculations with all required datasets are available as a supplement to this article. We suggest to preferably download *GloRiSe* 1.0 from the Zenodo-server, because it provides a DOI and a stable version. The github data storage serves for development purposes and will regularly be updated. Large updates, involving changes in the database structure will be noted by an integer number, while smaller updates (addition of samples) will be noted by the first digit. For reproducibility, we strongly encourage mention of *GloRiSe* version used. The MATLAB code used to produce the example presented in Section 5 and through which Figure 1 can be reproduced is available with detailed explanatory comments on the same page.

## 7 Conclusions

We introduce *GloRiSe*, a quasi-global database on river sediment composition including major, minor and trace elements along with nutrients and mineralogical data, placed in a spatio-temporal context by their metadata. This metadata also allows the user to trace back data sources, methods of preparation, measurement details and unit conversion, making the database comprehensible. The dataset is thought to enable global-scale investigation of geochemical fluxes related to erosion, weathering and sediment transport, to serve as a basis for statistical modelling and model validation, to screen promising basins for investigations or complement other datasets. With the database, we provide a MATLAB example script for the calculation of the sediment-flux weighted mean major element composition of riverine suspended sediment exported each year to the coastal oceans, We complement these estimates by an error analysis that accounts for the variability between basins and the uncertainty induced by limited knowledge about the relationship of sediment flux and its chemical composition.

## 8 Appendix

Units of solid concentrations were converted to weight % of sediment expressed as oxide, using molar masses and ratios in the following equation (2):

$$m(Oxide)_{wt\%} = (\frac{C(Element)_{mol\%}*M(Element)}{(\frac{M(Element)}{M(Oxide)})}) \tag{2}$$



C denotes concentrations in mol %, while M terms the molar mass (g/mol) and m is the mass percentage relative to the bulk
sediment. If concentrations were given relative to the solution volume (g/L or mol/L), they were only included if they were
convertible to weight % of sediment, i.e., when total suspended sediment concentration (TSS) was given:

$$m(Oxide)_{wt\% \, TSS} = \frac{m(Oxide)_{\frac{g}{L}}}{TSS_{\frac{g}{L}}} * 100 \tag{3}$$

Solution concentration are given in µmol/L and were converted to this using molar masses similar to equation (2). Original
units were noted for each entry and methods of measurement and sample treatment before measurement are specified as stated
in the reference to make the data and the conversion comprehensible and reproducible.

## 9 Team List

Gerrit Müller, g.muller@uu.nl, Department of Earth Sciences, Utrecht University, 3584 CB, Utrecht, The Netherlands

Jack J. Middelburg, j.b.m.middelburg@uu.nl, Department of Earth Sciences, Utrecht University, 3584 CB, Utrecht, The
Netherlands

Appy Sluijs, a.sluijs@uu.nl, Department of Earth Sciences, Utrecht University, 3584 CB, Utrecht, The Netherlands

## 10 Author contribution

GM structured the database, selected, collected and harmonized the data, created the evaluation examples and wrote the
manuscript. JM and AS provided advice on the nature of the data to be collected and feedback on the manuscript.

## 11 Competing interests

The authors declare that they have no conflict of interest.

## 12 Disclaimer

Despite quality control we cannot guarantee that no errors occurred during the initial data acquisition and publication or during
transcription into the database.

## 13 Acknowledgements

This work was funded by the Dutch Ministry of Education, Culture and Science through the Netherlands Earth System Science
Center (NESSC). AS thanks the European Research Council for Consolidator Grant 771497. We thank Eduardo A.F. Garzanti,



Dennis D. Eberl, Grace Andrews, Jérôme Viers and Volker Rachold for clarifying details and searching metadata related to their publications, that were included in the database. These researchers, along with Jens Hartmann, Gibran Romero-Mujalli and Stefan Kempe are furthermore thanked for literature suggestions. We thank Olivier J.T. Sulpis for advice regarding the
codes, and Gibran Romero-Mujalli for advice regarding database structure and compatibility. Svenja Trapp is thanked for testing user-friendliness of the database.

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
