# Peer review of "Introducing GloRiSe-A global database on river sediment composition"

_Earth System Science Data, 2021_

## Author Comment (AC1)

**Author response to Referee Comments on essd-2021-19**

Gerrit Müller, Jack J. Middelburg & Appy Sluijs

We thank the editor (Attila Demény), the invited reviewers (Thomas Gloaguen and Thorben Amann) and a voluntary data reviewer (Yutian Ke) not only for their constructive comments, but also for their kind words. A revised version of the manuscript will be submitted together with an updated version of the database, scripts and all assets including Table 1 and Figure 1. The updated database will be accessible as GloRiSe version 1.1 from Zenodo (doi: 10.5281/zenodo.4485795, Müller et al., 2021) and github (https://github.com/GerritMuller/GloRiSe).

**Reviewer #1 Thomas Gloaguen**

**General comment:**

The article and material is of very good quality and very useful, with consistent and comprehensive data. It provides a set of data of a global scope that does not exist so far, very important because referring to sediments and water from the main rivers in the world. In addition, it is an open source publication, which allows interaction with the help of any scientist in the world to expand the data set, seeming very promising. The source of the data, the different types of material and the methods used are very well described. References are appropriate. The dataset is of high quality, easy to download and use. The article is well written, with very few errors, and well structured. Some specific problems are described below. The data and the paper complement each other.

**Response:**

We gratefully thank Dr. Gloaguen for the helpful comments and kind support of our approach and data collection. Specific and detailed comments are addressed in the following:

**Specific comments:**

Line 96: about the sampling date. The authors mention "The closest information on sampling date is given for each observation. However, in the "SedimentDatabase\_ID" file, there are many samples with no date. This is the main problem of the dataset.

Response: "Samples without sampling date are those for which the original reference did not specify this, so that we are left without information. To limit this problem, we provide, as written in the cited line 96, "The closest information on sampling date [...] for each observation", e.g., season, month or year if only that was specified in the reference. Depending on the purpose, users may choose to apply the publication date instead, post-dating sampling and being available in the sheet 'SedimentDatabase\_ref'. We will retain our strategy, because we emphasize that this step should be taken with a lot of care and intentionally, as publication and sampling dates may be separated by several years" (Author Comment 1, https://doi.org/10.5194/essd-2021-19-CC1). However, to make this more clear, we added the following behind the original sentence in Line 96:

"However, many authors did not mention sampling date. In these cases, a user may choose to apply the publication date from the linked references (if appropriate), post-dating sampling ."

Line 150-177: I suggested reducing and summarizing (It would be more understandable).

Response: Following this advice, we summarized the description of uncertainty analysis and calculations of the sorbed pool as follows:

"Uncertainties are based on estimates of the temporal variability of sediment composition and fluxes (Appendix B). Estimates of the concentrations of major cations sorbed to negatively charged surfaces were derived using a published linear relationship between molar Al/Si ratios and cation exchange capacity (CEC), along with estimates of the average major element composition of this sorbed pool (Tipper et al., 2021)."

However, as a more detailed description is necessary to understand the presented numbers, we created Appendix B to accommodate the replaced description (former Line 150-177).

The section on sediment chemistry and CIX is a little too detailed for me. This contrasts with the lack of discussion about the chemical water data. I suggest a better balance in the discussion of the different data sets.

Response: "As *GloRiSe* is explicitly a database on river *sediment composition*, complementing existing databases on dissolved loads (e.g., GLORICH, Hartmann et al., 2014 or dissolved nutrients of McDowell et al., 2020). Therefore, solution composition is rather a little 'side-product' of *GloRiSe* and was only added if accompanying sediment composition. This will allow users to explore relations between particulate and dissolved loads. As it is not a major feature of *GloRiSe*, we do not want to draw too much attention on it by discussion in the text. Moreover, coverage of solution data is (for most elements) probably not sufficient to extract robust global averages or to discuss in detail. We wrote this in section '2 Data Collection' ("When instantaneous water discharge and/or suspended sediment concentration and/or solution properties [...] were reported in the same study, these were also added", Il. 89 – 91), but we will rephrase this in the revised version. Furthermore, we agree on that this section is too extensive and will shorten the discussion of CIX" (see Author Comment 1, https://doi.org/10.5194/essd-2021-19-CC1). We added the following phrase in Lines 91-92:

"However, the focus of data collection remains on the composition of solid phases and detailed information on water chemistry is available from other sources (Hartmann et al., 2014a; McDowell et al., 2020a; Virro et al., 2021)."

Moreover, we shortened the description and discussion of the CIX as follows: "Compared to the continental crust (Rudnick and Gao, 2013), Na2O, K2O and CaO are generally depleted in riverine suspended sediments, while Al2O3, Fe2O3 and TiO2 are enriched. This can be explained by weathering, transforming pristine mineral phases (e.g. feldspar, mafic minerals and calcite) into dissolved load and secondary phases (Putnis et al., 2014; Ruiz-Agudo et al., 2016). Wellsoluble elements (Na, K, Mg and Ca) will preferably be transported as dissolved load, leaving rather insoluble elements (Al, Fe, Ti) enriched in the secondary phase (Gaillardet et al., 1999; Garzanti et al., 2014a; Middelburg et al., 1988; Nesbitt and Young, 1982; Stroncik and Schmincke, 2001). Consequently, the unweathered source rock is relatively rich in mobile elements and also more reactive (Brantley et al., 2008; Lasaga, 1984). To quantify the relative contribution of weathered and unweathered material to fine-grained sediment samples, reflecting weathering intensity within the sediments source area, different chemical weathering indices have been developed (Fedo et al., 1995; Gaillardet et al., 1999; Garzanti et al., 2013, 2014b; Harnois, 1988; Nesbitt and Young, 1982; Parker, 1970). Most of them are based on the relative concentrations of mobile to guasi-insoluble elements (Fedo et al., 1995; Garzanti et al., 2014b; Harnois, 1988; Nesbitt and Young, 1982; Parker, 1970), each involving different elements and related pitfalls. For the current application, where we lack sufficient information on dissolved concentrations and do not expect a strong diagenetic imprint, but need to

account for carbonate- and phosphate-related CaO, we use the chemical index of alteration CIX (Garzanti et al., 2014a):

$$CIX = 100 * Al_2O_3 / (Al_2O_3 + Na_2O + K_2O)$$
(1)

We neglect the contributions of sorbed Na2O and K2O, because their magnitude is small compared to solid concentrations (Table 1). High CIX values imply a large contribution of weathered material, while a low CIX, similar to that of parent rocks, implies a substantial contribution of unweathered material. Therefore, our lower global average CIX values imply the exported material to be less weathered and hence more reactive than anticipated. Note, that the higher values from literature are not sediment-flux weighted. This observation is significant with respect to the propagated error of the CIX ( $\pm$  0.1 wt%) and is potentially explainable by the rivers included in each dataset: Cold-climate rivers exhibit lower chemical weathering rates in their catchment (Hartmann et al., 2014b) and mountainous rivers are characterized by steeper terrains and higher erosion rates, but less soil formation and chemical weathering (Milliman and Syvitski, 1992; Patton et al., 2018). This explanation is consistent with the marked increase of our sfw mean CIX compared to the mean and median estimates. The sfw mean is strongly influenced by a few large rivers draining areas of high chemical weathering intensities, i.e., Amazon, Ganga-Brahmaputra, Changjiang, Congo, Irrawady, Orinoco, Magdalena, Mekong and Godavari together already deliver ~ 20 % of the global sediment flux and South East Asian drainages contribute as much as 60 % of the global sediment budget (Milliman and Farnsworth, 2011)."

The quality of Figure 1 should be improved.

Response: We inserted a new, high quality figure as a Robinson projection, being also a more accurate representation in terms of areas/sizes and giving an impression of the rivers sizes and fluxes by indication of discharge from *HydroBasins* (Linke et al., 2019). The figure will also be available as a separate high quality file for publication. The following figure was inserted:

Figure 1 Spatial extent of the database. Dot size indicate the number of samples available per location. Dot colour indicates what kind of data is available at that location (ME: Major & minor elements, TE: Trace elements, Min: Mineralogical or petrological composition). River water discharge, as indicated by blue coloration, was taken from the *HydroBasins* database at Pfaffstetter level 7 (Linke et al., 2019).

**Detailed comments:**

Line 92: let clear what is Appendix A. Are there other appendices? (not found)

Response: Appendix B was newly created to accommodate the detailed descriptions of uncertainty analysis and estimation of the sorbed loads (see above).

Line 93: it would be better to standardize using only coordinates, without referring to

Google Earth or maps (retrieving the coordinates for all locations)

Response: We removed the reference to Google Earth.

Line 94: What are the 21 samples?

Response: We inserted "[...], denoted 'sa' as the 'observation type'" behind in Line 94 to give the key to identify those samples.

Line 181-182: review the sentence

Response: We rephrased as follows: "Most of the compilation of Viers et al., (2009) and the data of Martin and Meybeck (1979) were incorporated into *GloRiSe.*"

Line 225: "T denotes the total concentration" – remove the parentheses - or complete the sentence

Response: We completed the sentence as follows: "T denotes the total concentration of ferric + ferrous Fe or organic + inorganic P, respectively."

Line 248. Substitute comma by point.

Response: We changed as suggested.

**Reviewer #2 Thorben Amann**

**General comment:**

The manuscript presents a data compilation of the composition of suspended solids in rivers. The authors have collected a comprehensive set of data, which covers almost all regions of the world. This should enable a plethora of new studies on sediment transport to the oceans, riverine biogeochemical cycling, weathering fluxes, and many more. The manuscript is timely fills a much needed gap for global studies. I recommend the publication after minor revisions.

I have one general point, which I stumbled upon: The title says the database is about sediments, and then it is actually only about sediments (L71: "Riverbed sediments "), where no data on suspended matter (L68: "suspended sediment") was available. While I understand the differentiation, I find it a bit confusing when I just read the title. I am not concerned about this, I just want to point out, that there is potential for misunderstandings, which may be resolved by a slight change of the used terms. I leave the decision to the authors.

Response: Dr. Amann is thanked for his careful, constructive assessment of both, the data and the manuscript, and for providing the useful tool for exploratory data analyses. Regarding the title, we have internally discussed which terminus would be best understandable before the first submission, because different terminology is currently used in different disciplines (e.g., Geochemists tend to use 'particles' in relation to suspended sediment, while Geomorphologists tend to use 'sediment' when referring to either riverbed sediments or suspended sediments or both). We wanted to stress that *GloRiSe* includes both types, riverbed sediment and suspended sediment, and chose to consistently use 'sediment', when referring to both, while we explicitly state 'suspended sediment' or 'riverbed sediment', when referring to one of these types. The term 'sediment' appeared to be the most general and potentially best known by the interdisciplinary readership. *GloRiSe* is open to include more data on bed sediments in future, depending on desired applications. The structure of the database allows for easy integration and differentiation of those sample types.

**Following the step-by-step guideline for reviewers:**

Are the data and methods presented new? Yes. To my knowledge, there is no other comprehensive data compilation on river suspended matter.

Is there any potential of the data being useful in the future? Definitely, as stated above.

Are methods and materials described in sufficient detail? Are any references/citations to other data sets or articles missing or inappropriate? Everything is well described and comprehensible.

Is the article itself appropriate to support the publication of a data set? Yes

Check the data quality: is the data set accessible via the given identifier? Yes.

Is the data set complete?

Yes, with the limitations described in the MS itself (Section 4)

Are error estimates and sources of errors given (and discussed in the article)? Are the accuracy, calibration, processing, etc. state of the art? Are common standards used for comparison?

This doesn't really apply here. But shortcomings or problems merging different data sources into one comprehensive database were discussed in the MS.

Is the data set significant - unique, useful, and complete?

Consider article and data set: are there any inconsistencies within these, implausible assertions or data, or noticeable problems which would suggest the data are erroneous (or worse). If possible, apply tests (e.g. statistics). Unusual formats or other circumstances which impede such tests in your discipline may raise suspicion.

Although I didn't experience any problems using the dataset with Python/Pandas, it may be advisable to change the headers to names containing no special characters (like  $\mu$ , a dash, a smaller than sign...). Programs like ArcGIS, for example, do not like those characters in the header.

Response: Thanks for reminding us about this. We changed the headers to exclude such special characters. For instance, ' $\mu$ ' was replaced by 'mu', dashes were substituted by underscores or removed, smaller/greater than signs were removed, 'wt%' was changed to 'wt' and other units including '%' were substituted by '\_perc'.

Generally, in all files:

The use of spaces, underscores or no space between parameter and units is not consistent. Units are also not given consistently (could be derived from documentation, but better give unit in header). I recommend a second screening of the headers to unify the appearance. Response: We removed spaces and hyphens, also because of the above comment on special characters, and now use underscores consistently as follows: measured property\_additional info\_unit. Additional information is optional and can refer to organic or inorganic. Unit can consist of several blocks, e.g., mg\_L for mg/L.

Overall, I found some inconsistencies in the data using Python and the package Pandas Profiling (https://github.com/pandas-profiling/pandas-profiling). I will point out some found issues here, but strongly recommend looking into a tool for exploratory data analyses (another recommendation: https://github.com/sfu-db/dataprep) to find flaws in the dataset that relate to format, data types or other formal issues.

Response: We thank Dr. Amann a lot for this suggestion and screened for more errors (see Additional Comments at the end of this document). However, as the tool is applicable only in Python and we used MATLAB for all data processing, the single operations described in the 'README.md' of the suggested tool (https://github.com/pandas-profiling/pandas-profiling#readme) were executed manually in MATLAB (see Additional Comments at the end of this document).

Specific points (very selective, there may be more):

SedimentDatabase\_ME\_Nut.csv

No Sample\_ID/Location\_ID/SeaCat/Observation/type/Sampletype/Basin\_ID/Original\_UnitME/Treatment/Method from line 2411 to end, maybe I missed an explanation, but as without identifier, the data is rather useless, isn't it?

Response: This is indeed a leftover from earlier corrections, where we forgot to remove temporarily placed data. These lines were removed now, because they are already incorporated at the correct place with correct IDs ("USA-MMY-CLK[NUMBER]").

The csv ends with 1 useless (empty columns)

Response: This column was removed.

Column "filter size\_>µm" contains 88 values named "cent". Is this correct?

Response: This is correct. 'cent' refers to centrifuged, with no numeric information on grain size. We rather give this as a keyword, than not giving any information. All keywords for each column are listed in the second sheet of the excel files, together with units and abbreviations. This table is added to the documentation for better visibility.

SedimentDatabase\_Minerals.csv Header: tottal mafic – remove 't'

Response: We changed this into 'totalMafic'.

The csv ends with 4 useless (empty columns)

Response: These columns were removed.

8 Filter size (> μm) 9 Sieve size ((<) μm) -- inconsistent use of parenthesis

Response: This was addressed when changing special characters.

The abbreviations used in the headers should be written out in a table in the Documentation

Response: We copied the table with headers and explanations from the second sheet of the corresponding excel file into the documentation.

SedimentDatabase\_TE.csv

Column Be\_ppm and maybe others contain values with a less-then sign. This may be correct to report like this but makes it hard to process the data with software like Python, Matlab et al, as they will handle the entire column data as string or object. The user then has to manually find and replace the values.

Response: We substituted these values by half the indicated detection limit (value behind the smaller than sign).

**Is the data set itself of high quality?**

Check the presentation quality: is the data set usable in its current format and size? Are the formal metadata appropriate? Check the publication: is the length of the article appropriate? Is the overall structure of the article well-structured and clear? Is the language consistent and precise? Are mathematical formulae, symbols, abbreviations, and units correctly defined and used? Are figures and tables correct and of high quality? Everything looks well suited for publication.

The Figure 1 quality should be improved.

**Response: We changed the figure, see response to Reviewer #1.**

I think the exemplary part (Section 5) is very extensive and goes into a whole lot of detail. I feel this could be shortened a bit (but not left out, just maybe moved to the appendix), to focus on the results that can be achieved.

Response: By explaining the procedures in detail, the reader/user may get a better impression of which possibilities are available to select, group and treat samples. As this metadata is an important part of *GloRiSe*, we left the section within the main text. We did not aim to focus on the results only, but also on how to achieve and interpret them using the provided metadata. However, we see the need to be more concise at this point in the main text, so we created Appendix B to accommodate descriptions of the uncertainty analysis and of the estimation of the sorbed pool. The subsequent discussion of the results and CIX was also shortened (see response to *Reviewer #1*).

Also, it could be nice to have an overview table with the parameters included in the database, together with the basic statistics like count, mean, median, min, max, percentiles...

Response: We give descriptive statistics (count, mean, median, sediment flux-weighted mean, standard deviation) in Table 1 for the basin-wise averages. Counts for each location are given in the file "SedimentDatabase\_Locations". Statistics from unprocessed bulk data (e.g., location-wise averages for each variable) would result in meaningless or at least heavily biased numbers, because it

would mostly depend on the sample selection in the database and sampling frequencies needed for the original study. Therefore, we refrain from presenting this kind of data.

Is the data set publication, as submitted, of high quality? Yes, from my perspective it looks well put together. Just the data itself needs some more cleaning/trimming, as described above.

**Response: See Additional Comments at the end of this document.**

Finally: By reading the article and downloading the data set, would you be able to understand and (re-)use the data set in the future? Yes.

**Specific comments:**

89ff Maybe good to reference specific databases like the Glorich or the new GRQA (under review in the same journal: https://doi.org/10.5194/essd-2021-51)

Response: We added the following: "However, the focus of data collection remains on the composition of solid phases and detailed information on water chemistry is available from other sources (Hartmann et al., 2014a; McDowell et al., 2020a; Virro et al., 2021)."

92 "Conversed" if converted & 92 Define "properly". What is the reasoning behind the conversion? I guess there is no one doing it not properly on purpose.

Response: We rephrased as follows: "Unit conversion is detailed in the Appendix." We did not mean to claim errors in literature, but just wanted to say, that we harmonized the data in terms of units (e.g.,  $\mu$ mol/L to wt% of dry sediment) and the used equations can be found in Appendix A.

103 How were the basins identified?

Response: We added: "[...] and was assessed by visually tracing the streams to their mouth (using © Google Earth 2020)".

108 How was this done with Google Earth? Visually?

Response: See above.

125 The figure deserves a higher resolution to avoid compression artefacts.

Response: We changed the figure, see above and response to Reviewer #1.

139f What was the reason for the decision to exclude coarser grainsizes? I imagine it is because you assume they don't get suspended, but it would be good to have a short explanation on the criterion.

Response: This is correct, we added "[...] to represent only suspended sediment".

**Additional Comments**

**Voluntary Reviewer Yutian Ke**

We thank Dr. Yutian Ke for his helpful comments that he sent us without request. He pointed out the following minor errors in the database:

RiverNames.xlsx

A157 "Irrawady" should be "Irrawaddy";

SedimentDatabase\_ME\_Nut.xlxs

Wrong number W1608 X1933 X1949 Z447;

Strange number Row 1935 etc., small number;

Sample type, what is "BS/BS"?

SedimentDatabase\_TE.xlsx

Wrong number Y80/85/92 AB94 AI79/87 AC89/95 AH85 AV964 AX960

Strange number Q905 BA1636 BD1651 BH1653 BL1653, large number.

Response: We identified the errors by comparison with the original data and reference and corrected them.

**The authors**

Following the suggestions of Reviewer #2, we applied some tests (histograms and descriptive statistics, outliers, unique and most common values) and found some more errors, that were corrected. These are:

"SedimentDatabase\_Minerals"

Conversion of fraction to percentage (II. 81 - 93; AO104 & AP104)

"SedimentDatabase\_ME\_Nut"

Unit conversion (µmol/g -> µmol/L) (AK1886-AT1920; AK2082 - AR2182)

**References**

[revised manuscript text omitted]